# Auxiliary Reference Samples for Extrapolating Spectral Reflectance from Camera RGB Signals

**DOI:** 10.3390/s22134923

**Published:** 2022-06-29

**Authors:** Yu-Che Wen, Senfar Wen, Long Hsu, Sien Chi

**Affiliations:** 1Department of Electrophysics, National Yang Ming Chiao Tung University, No. 1001 University Road, Hsinchu 300, Taiwan; adam0682009.ep06g@nctu.edu.tw (Y.-C.W.); long@nctu.edu.tw (L.H.); 2Department of Electrical Engineering, Yuan Ze University, No. 135 Yuan-Tung Road, Taoyuan 320, Taiwan; 3Department of Photonics, National Yang Ming Chiao Tung University, No. 1001 University Road, Hsinchu 300, Taiwan; schi@mail.nctu.edu.tw

**Keywords:** spectrum reconstruction, spectral reflectance recovery, linear interpolation, principal component analysis

## Abstract

Surface spectral reflectance is useful for color reproduction. In this study, the reconstruction of spectral reflectance using a conventional camera was investigated. The spectrum reconstruction error could be reduced by interpolating camera RGB signals, in contrast to methods based on basis spectra, such as principal component analysis (PCA). The disadvantage of the interpolation method is that it cannot interpolate samples outside the convex hull of reference samples in the RGB signal space. An interpolation method utilizing auxiliary reference samples (ARSs) to extrapolate the outside samples is proposed in this paper. The ARSs were created using reference samples and color filters. The convex hull of the reference samples and ARSs was expanded to enclose outside samples for extrapolation. A commercially available camera was taken as an example. The results show that with the proposed method, the extrapolation error was smaller than that of the computationally time-consuming weighted PCA method. A low cost and fast detection speed for spectral reflectance recovery can be achieved using a conventional camera.

## 1. Introduction

Surface spectral reflectance is useful for the color reproduction of industrial products and artwork [1,2,3]. It can be measured directly with an imaging spectrometer [4,5]. However, direct spectral measurements are expensive. Indirect measurements using the spectrum reconstruction technique are of interest [6,7,8,9,10,11,12,13,14]. The spectrum of the image pixel is reconstructed from the channel outputs of the image acquisition device. Since no diffractive optical imaging system is required, the indirect method has the advantages of a low cost and fast detection speed. Therefore, using a conventional camera makes more field applications possible, e.g., smartphone cameras used as sensors to measure surface spectral reflectance.

Orthogonal projection [6], principal component analysis (PCA) [7,8], Gaussian mixture [9], non-negative matrix transformation (NMT) [10,11] and interpolation [11,12,13,14] have been proposed for spectrum reconstruction. Indirect methods that require training spectra are also known as learning-based methods, such as orthogonal projection, PCA and NMT. The training spectra are used to derive basis spectra. The reconstructed spectrum is a linear combination of basis spectra. The coefficients of basis spectra can be solved from simultaneous equations describing the channel outputs of the imaging device. The accuracy of the reconstructed spectrum increases with the number of channels. For cases with a conventional tricolor camera, where only three channels are available, the accuracy of the reconstructed spectrum might not be high enough.

The interpolation method uses reference spectra to reconstruct a spectrum interpolated from input values, e.g., XYZ tristimulus values [11,12,13] and RGB signal values [14]. Due to the use of a look-up table (LUT) to store the reference spectra, this method is often referred to as the LUT method. The authors of [11,12,13,14] showed that the LUT method has the advantage of being more accurate than the PCA method, where the reference spectra for interpolation are the same as the training spectra for the PCA method. A spectrum is interpolated from its neighboring reference samples using the LUT method, whereas the basis spectra of the PCA method are derived from all training samples. The weighted PCA (wPCA) method was proposed to enhance the contribution of neighboring training samples in the CIELAB color space to the basis spectra [8,14]. Since basis spectra depend on the sample to be reconstructed, the computation time of the wPCA method is significantly increased compared to the conventional PCA method and the LUT method.

Learning-based methods require camera spectral sensitivities to formulate simultaneous equations describing the channel outputs. Camera spectral sensitivities can be directly measured using a monochromator [15], but accurate measurement is expensive. Without the use of a monochromator, the camera spectral sensitivities can be estimated by solving a quadric minimization problem [15,16]. An alternative approach is to estimate the spectral sensitivities including the camera and light source so that the reflectance spectrum can be calculated from the camera signals [17,18,19,20,21]. The estimation errors of spectral sensitivities cause additional errors in the reconstructed spectrum.

The LUT method does not require the spectral sensitivity functions because the reconstructed spectrum is interpolated from the measured spectra of the reference samples. However, if the sample lies outside the convex hull of the reference samples in the RGB signal space, it cannot be interpolated. Such a sample can be called an outside sample to distinguish it from the samples inside the convex hull. In the literature, modified PCA and NMT methods have been used to extrapolate outside samples [11,12,13,14], although spectral sensitivity functions are required. The authors of [11,12,13] considered interpolation in the XYZ color space, where spectral sensitivity functions were equivalently assumed to be the CIE color matching functions (CMFs). The authors of [14] considered interpolation in the RGB signal space, where the camera was assumed to follow the sRGB standard so that RGB signal values and XYZ tristimulus values can be converted to each other via the well-known sRGB matrix. This hypothetical camera is called the sRGB camera, and its spectral sensitivities are presented in [22].

The authors of [11,13] used 2D interpolation and 3D interpolation to extrapolate outside samples from reference samples, respectively. It is not guaranteed that the 2D interpolation method will extrapolate all outside samples [11]. The authors of [14] extrapolated outside samples from reference samples and additional reference samples; the latter are called model-based metameric spectra of extreme points (MMSEPs). The extreme points are the eight corners of the RGB signal cube. They are black, white, red, green, blue, yellow, cyan and magenta, corresponding to the signal vectors [R, G, B]^T^ = [0, 0, 0]^T^, [1, 1, 1]^T^, [1, 0, 0]^T^, [0, 1, 0]^T^, [0, 0, 1]^T^, [1, 1, 0]^T^, [0, 1, 1]^T^ and [1, 0, 1]^T^, respectively, where the maximum values of the signals are normalized to 1.0; the subscript T denotes the transpose operation. The metameric spectra are the reflection spectra from eight surfaces under D65 illumination. The spectral reflectance of the eight surfaces was constructed using the sRGB camera. The MMSEPs were equivalently constructed using the spectral sensitivities of the sRGB camera.

Inspired by [14], we propose the use of auxiliary reference samples (ARSs) for extrapolating outside samples using the LUT method. ARSs are high-saturation samples. They are created using appropriately chosen color filters and color chips. Color filters are in turn mounted on the spectroradiometer to measure the spectrum of filtered reflection light from a color chip. The RGB signal values corresponding to the filtered reflection light are recorded by a camera mounted with the same color filter. Color filters and color chips are chosen so that outside samples can be enclosed by reference samples and ARSs in the RGB signal space for extrapolation. Numerical studies of the proposed method were carried out. A comparison of the LUT method utilizing ARSs, the LUT method utilizing MMSEP samples, the wPCA method and other methods is presented. For ease of reference, Table 1 lists the abbreviations defined herein in alphabetical order.

## 2. Materials and Assessment Metrics

A Nikon D5100 camera was taken as an example. Its spectral sensitivities of the red, green and blue signal channels measured by a monochromator are shown in Figure 1a [16]. The average wavelengths of the spectral sensitivities of the red, green and blue channels are denoted as *λ*_CamR_, *λ*_CamG_ and *λ*_CamB_, respectively, which are called the channel wavelengths for simplicity. The full width at half maximum (FWHM) of the spectral sensitivities of the red, green and blue channels is denoted as Δ*λ*_CamR_, Δ*λ*_CamG_ and Δ*λ*_CamB_, respectively. The spectral specifications of the camera are shown in Table 2.

The reflectance spectra of matt Munsell color chips measured by a Perkin-Elmer lambda 9 spectroradiometer were adopted for preparing reference samples and test samples [23]. The available measurement data in [23] comprise 1269 records, but 2 of them are duplicates, namely, record 1242 (annotation 10RP 7/2) and record 1249 (annotation 10RP 7/4). Therefore, 1268 reflectance spectra were used in this paper. The light source was assumed to be illuminant D65. A total of 202 color chips were selected for the preparation of the reference samples.

A spectrum can be represented by the vector ***S*** = [*S*(*λ*_1_), *S*(*λ*_2_), …, *S*(*λ_M_*_w_) ]^T^, where *S*(*λ_j_*) is the spectral amplitude at wavelength *λ_j_*, *λ_j_* = *λ*_1_ + (*j* − 1)Δ*λ* is the *j*-th sampling wavelength, *j* = 1, 2, …, *M*_w_, and Δ*λ* is the wavelength sampling interval; *M*_w_ is the number of sampling wavelengths. In this paper, spectra were sampled from 400 nm to 700 nm in steps of 10 nm, i.e., *λ*_1_ = 400 nm, Δ*λ* = 10 nm and *M*_w_ = 31. The spectrum vector of the light reflected from a color chip is ***S***_Reflection_ = ***S***_Ref_
∘
***S***_D65_, where ***S***_Ref_ and ***S***_D65_ are the spectral reflectance vector of the color chip and the spectrum vector of the illuminant D65, respectively; the operator ∘ is the Hadamard product, also known as the element-wise product. Figure 2a shows the color points of the reflection light from the 1268 Munsell color chips in the CIELAB color space, where the 202 reference samples and 1066 test samples are shown as red and blue dots, respectively. Figure 2b–d are the same as Figure 2a, but with different viewing angles. The CIE 1931 CMFs were adopted in this paper.

The measured signal of a color channel is *U*_Meas_ = ***S***_Reflection_^T^***S***_U_, where *U* = *R*, *G* and *B* for the red, green and blue channels, respectively; ***S***_Reflection_ is the reflection spectrum vector; ***S***_U_ is the spectral sensitivity of the channel. For the white balance condition, the channel signals are normalized to *U* = *U*_Meas_/*U*_MeasD65_, where *U* = *R*, *G* and *B*; *U*_MeasD65_ is the measured signal when ***S***_Reflection_ = ***S***_White_ ∘
***S***_D65_; ***S***_White_ is the spectral reflectance of a white card. The white side of a Kodak gray card was taken as the white card, where its spectral reflectance is approximately 0.9 in the visible wavelength range. The vector representing the camera signals is designated as ***C*** = [*R*, *G*, *B*]^T^. Figure 3a shows the color points of the reflection spectra from the Munsell color chips in the RGB signal space using the Nikon D5100, where the 202 reference samples and 1066 test samples are shown as red and blue dots, respectively. There are 62 samples in the convex hull of the 202 reference samples. Figure 3b shows the convex hull.

For a given test signal vector, the LUT method to reconstruct the reflection light spectrum is shown in Section 3.1. The reconstructed spectrum vector is designated as ***S***_Rec_. The reconstructed spectral reflectance vector ***S***_RefRec_ was calculated as the reflection spectrum vector ***S***_Rec_ divided by the D65 spectrum vector ***S***_D65_ element by element.

The reconstructed spectral reflectance vector ***S***_RefRec_ was assessed by the root mean square (RMS) error *E*_Ref_
*=* (|***S***_RefRec_ − ***S***_Ref_|^2^/*M*_w_)^1/2^ and the goodness-of-fit coefficient *GFC* = |***S***_RefRec_^T^***S***_Ref_|/|***S***_RefRec_| |***S***_Ref_|, where |·| stands for the norm operation. The color difference between ***S***_Rec_ and ***S***_Reflection_ was assessed using CIEDE2000 Δ*E*_00_. The spectral comparison index (*SCI*) was also used to assess the reconstructed results [24,25]. The parameter *k* in the formula for calculating *SCI* shown in [24] was set to 1.0. For the values of *E*_Ref_, Δ*E*_00_ and *SCI*, the smaller, the better. The statistics of the three metrics were calculated, which are the mean *μ*, standard deviation *σ*, 50th percentile *PC50*, 98th percentile *PC98* and maximum *MAX*. For the value of *GFC*, the larger, the better. The statistics of *GFC* were calculated, which are the mean *μ*, standard deviation *σ*, 50th percentile *PC50* and minimum *MIN*. The fit of the spectral curve shape is good if *GFC* > 0.99 [14,26]. The ratio of samples with *GFC* > 0.99 was calculated, which is called the ratio of good fit and designated as *RGF99*.

## 3. Spectrum Reconstruction Method

### 3.1. Reflection Spectrum Reconstruction

This subsection describes the LUT method to reconstruct the reflection spectrum vector ***S***_Rec_ from the test signal vector ***C*** [12]. The color points of the reference samples in the RGB signal space were not regularly distributed as shown in Figure 3a. Therefore, the use of the scattered data interpolation method was required. Several interpolation methods were surveyed in [27]. Among these methods, linear tetrahedral interpolation was adopted due to its simplicity and computational time savings [11,12,13]. A tetrahedral mesh in the RGB signal space was generated from the reference signal vectors. Note that the tetrahedrization is not unique, and the interpolation result depends on the tetrahedrization [12,27]. All programs in this paper were implemented in MATLAB (version R2021a, MathWorks). The tetrahedral mesh used for interpolation was generated by the MATLAB function “delaunay” [11,13,14]. There were three steps to interpolate the test sample.

STEP 1: Locate the tetrahedron.

The tetrahedron that encloses the color point Q of the vector ***C*** in the RGB signal space was located. Figure 4 shows the tetrahedron, with vertices Q_1_, Q_2_, Q_3_ and Q_4_ enclosing the color point Q. A database or look-up table storing the tetrahedral mesh can be used to save processing time in locating the tetrahedron [13]. This paper used the MATLAB function “pointLocation” to locate the tetrahedron, which is a related function of “delaunay”.

STEP 2: Calculate interpolation coefficients.

The reference signal vectors of Q_1_, Q_2_, Q_3_ and Q_4_ were assumed to be ***C***_1_, ***C***_2_, ***C***_3_ and ***C***_4_, respectively. It is required that ***C*** is the linear combination of the reference signal vectors, and
***C*** = *α*_1_***C***_1_ + *α*_2_***C***_2_ + *α*_3_***C***_3_ + *α*_4_***C***_4_,(1a)
1 = *α*_1_ + *α*_2_ + *α*_3_ + *α*_4_,(1b)
where the coefficients *α*_1_, *α*_2_, *α*_3_ and *α*_4_ are weighting factors. Equation (1a) comprises three scalar equations because tristimulus vectors are 3D. Equation (1b) guarantees that Q is inside the tetrahedron if 0 < *α*_1_, *α*_2_, *α*_3_, *α*_4_ < 1. The four coefficients in Equation (1a,b) were solved.

STEP 3: Calculate the reconstructed reflection spectrum.

The reconstructed reflection spectrum vector is
***S***_Rec_ = *α*_1_***S***_1_ + *α*_2_***S***_2_ + *α*_3_***S***_3_ + *α*_4_***S***_4_,(2)
where ***S****_j_* is the reference spectrum vector corresponding to the vertex Q*_j_* for *j* = 1, 2, 3 and 4. If the reconstructed spectrum has negative values, the value is set to zero.

If both sides of Equation (2) are multiplied by the spectral sensitivity function of a signal channel and integrated over the wavelength, we obtain Equation (1a) corresponding to the signal channel. However, the interpolation is an inverse problem. The reconstructed spectrum vector ***S***_Rec_ is one of numerous metameric spectrum vectors corresponding to the test signal vector ***C***. Equation (1a,b) are the four constraints for finding a metameric spectrum vector. The difference between the target spectral reflectance vector ***S***_Ref_ and the reconstructed spectral reflectance vector ***S***_RefRec_ calculated from the metameric spectrum vector ***S***_Rec_ was assessed using the metrics defined in Section 2.

### 3.2. Spectral Reflectance Reconstruction Workflow

Figure 5 shows a flow chart for reconstructing the spectral reflectance vector ***S***_RefRec_ from the test signal vector ***C***. The convex hull of the tetrahedral mesh of the reference signal vectors is denoted as H_R_. An example of H_R_ is shown in Figure 3b. The convex hull of the tetrahedral mesh of the reference signal vectors and ARS vectors is denoted as H_RA_. The method for creating ARSs is shown in Section 4.

If the test signal vector is inside H_R_, its reflection spectrum vector ***S***_Rec_ is interpolated from the reference samples using the three-step procedure in Section 3.1. If the test signal vector is outside H_R_ and inside H_RA_, its reflection spectrum vector ***S***_Rec_ is extrapolated from the expanded reference sample set including the reference samples and ARSs using the three-step procedure in Section 3.1. If the test signal vector is outside H_RA_, its reflection spectrum vector must be extrapolated using the other method. Therefore, ARSs must be chosen to guarantee that the test signal vectors of interest are inside H_RA_.

## 4. Auxiliary Reference Samples

### 4.1. ARS Creation

The ARSs were described in the last paragraph of Section 1. Figure 6 shows a five-step flow chart for creating a set of ARSs. The description below uses the Nikon D5100 as an example. The example color filters were optimized for the Nikon D5100.

STEP 1: Select reflective surfaces.

The Munsell color chips in the convex hull in CIELAB and the white card were used as reflective surfaces to create ARSs. The number of color chips in the convex hull in Figure 2a is *N* = 62. The reference samples in the convex hull in Figure 3b are the reflection samples from the same 62 color chips. Samples of the white point and black point are default ARSs. The white ARS is the white point sample, whose signal vector is [1, 1, 1]^T^. The illuminant is D65. The spectrum of the black ARS is zero, and its signal vector is [0, 0, 0]^T^.

STEP 2: Select color filters.

Appropriate cyan, yellow and magenta filters were selected. They were used to filter reflection light to increase color saturation. Figure 7a shows the spectral transmittance of an example filter set. Given a reference sample of a signal vector [*R*, *G*, *B*]^T^, its signal vector becomes [*R*_f_, *G*_f_, *B*_f_]^T^ after filtering. If the filter is cyan, the ratios *B*_f_/*R*_f_ and *G*_f_/*R*_f_ will be larger than *B*/*R* and *G*/*R*, respectively, and a more saturated sample is created. If the reference sample is magenta (*G* << *B*, *R*), a highly saturated blue sample is created. The issue of color filter selection is discussed further in Section 4.2.

STEP 3: Measure raw ARSs.

The color filters selected in STEP 2 were sequentially mounted on the camera and the spectroradiometer for measurement. For each color filter, the RGB signal values and spectrum of the reflection light from the reflective surfaces selected in STEP 1 were measured. There were 3*N* + 3 = 189 measured samples using the color filters, called the raw ARSs. In this paper, the RGB signal values and spectra were calculated according to Section 2 for numerical study. Figure 8a shows an example of the raw ARSs in the RGB signal space, where the color filters in Figure 7a are used. In Figure 8a, the raw ARSs from the color chips and the white card are shown as red dots and crosses, respectively.

STEP 4: Create amplified raw ARSs.

From Figure 3a and Figure 8a, we can see that the 3*N* raw ARSs from the color chips cannot properly enclose the test samples due to attenuation of the reflection light passing through the filter. The spectra and RGB signal vectors of these raw ARSs were multiplied by an amplification factor γ greater than 1.0. If an RGB signal vector is multiplied by an amplification factor, its color point in the RGB signal space moves away from the black point in the direction from the black point to its original color point. The value of γ could be sample-dependent to reduce extrapolation error. For simplicity, γ = 1.5 was empirically set for all these samples except for the samples corresponding to the color chips of L* = 90 in Figure 2a–d, which have a value of 9 in the Munsell annotation. The exception samples were multiplied by a smaller amplification factor so that they remained within the RGB signal cube, where γ = 1.185. All amplified raw ARSs are shown as blue dots in Figure 8a.

STEP 5: Select the ARSs.

The ARSs are the samples in the convex hull of the amplified raw ARSs, the three raw ARSs from the white card, the white ARS and the black ARS. The latter two are shown as green crosses in Figure 8a. The convex hull is denoted as H_A_ and is shown as a blue mesh in Figure 8b. The total number of ARSs in H_A_ is 53. The convex hull of the reference samples H_R_ defined in Section 3.2 is also shown as a red mesh in Figure 8b for comparison. From Figure 8b, H_R_ is completely inside H_A_. In this case, H_A_ and H_RA_ are the same because the color points of the reference samples are well enclosed by H_A_. It seems unnecessary to measure 189 samples in STEP 3 as there are only 53 samples in H_A_. However, before building H_A_, we do not know which samples will be in H_A_.

### 4.2. Color Filter Design Method

Only cyan, yellow and magenta filters were selected in STEP 2 and used in STEP 3. Extrapolation error can be further reduced by using more color filters, e.g., using additional red, green and blue filters. This paper limited the number of color filters to three because (1) the use of cyan, yellow and magenta color filters enables all outside samples to be extrapolated for the cases under consideration, and (2) the cost of creating ARSs increases with the number of color filters. The spectral transmittance functions of the considered cyan, yellow and magenta filters are based on a super-Gaussian function. Such color filters can be absorption filters or interference filters [28]. There are stock color filters in various specifications on the market.

The cyan filter is a short-pass optical filter whose spectral transmittance is assumed to be
(3)$$f_{C}\left(\lambda \right)=f_{C0}\exp\left[-{\left(\frac{\lambda -\lambda_{S}}{\sigma_{C}}\right)}^{a_{C}}\right]$$
where *f_C_*_0_ is the maximum transmittance; *λ**_S_* = 400 nm; σ*_C_* and *a_C_* are parameters determined by the filter edge wavelength *λ**_C_* and edge width Δ*λ**_C_*. Figure 9a shows the definitions of *λ**_C_* and Δ*λ**_C_*. The edge wavelength *λ**_C_* is the wavelength of the half-maximum transmittance, i.e., *f*_C_(*λ**_C_*) = 0.5 *f_C_*_0_. The edge width Δ*λ**_C_* is the wavelength interval from 0.1 *f_Y_*_0_ to 0.9 *f_Y_*_0_. From Equation (3) and the definitions of *λ**_C_* and Δ*λ**_C_*, the following equation can be derived.
(4)$$\left[{\left(\ln10\right)}^{\frac{1}{a_{C}}}-{\left(-\ln0.9\right)}^{\frac{1}{a_{C}}}\right]\left(\lambda_{C}-\lambda_{S}\right)={\left(\ln2\right)}^{\frac{1}{a_{C}}}\Delta \lambda_{C},$$

Given the values of *λ_C_* and Δ*λ_C_*, the value of *a_C_* can be solved from Equation (4) using the MATLAB function “fzero”. After the value of *a_C_* is solved, the parameter σ*_C_* can be easily calculated by
(5)$$\sigma_{C}={\left(\ln2\right)}^{-\frac{1}{a_{C}}}\left(\lambda_{C}-\lambda_{S}\right)$$

The yellow filter is a long-pass color filter whose spectral transmittance is assumed to be
(6)$$f_{Y}\left(\lambda \right)=f_{Y0}\exp\left[-{\left(\frac{\lambda_{L}-\lambda }{\sigma_{Y}}\right)}^{a_{Y}}\right]$$
where *f_Y_*_0_ is the maximum transmittance; *λ**_L_* = 700 nm; σ*_Y_* and *a_Y_* are parameters determined by the filter edge wavelength *λ**_Y_* and edge width Δ*λ**_Y_*. Figure 9a also shows the definitions of *λ**_Y_* and Δ*λ**_Y_*, which are similar to those of *λ**_C_* and Δ*λ**_C_*. The parameters σ*_Y_* and *a_Y_* can be solved from the same equations as Equations (4) and (5), except that *a_C_*, (*λ**_C_* − *λ**_S_*) and Δ*λ**_C_* in Equation (4) are replaced by *a_Y_*, (*λ**_L_* − *λ**_Y_*) and Δ*λ**_Y_*, respectively, and σ*_C_*, *a_C_* and (*λ**_C_* − *λ**_S_*) in Equation (5) are replaced by σ*_Y_*, *a_Y_* and (*λ**_L_* − *λ**_Y_*), respectively.

The magenta filter is a notch optical filter whose spectral transmittance is assumed to be
(7)$$f_{M}\left(\lambda \right)=f_{M0}\left(1-\exp\left[-{\left|\frac{\lambda -\lambda_{M}}{\sigma_{M}}\right|}^{a_{M}}\right]\right)$$
where *f_M_*_0_ is the maximum transmittance; *λ**_M_* is the central wavelength; σ*_M_* and *a_M_* are parameters determined by the wavelength separation Δ*λ**_Sep_* and edge width Δ*λ**_M_*. Figure 9b shows the definitions of Δ*λ**_Sep_* and Δ*λ**_M_*. The wavelength separation Δ*λ**_Sep_* = *λ**_ML_* − *λ**_MS_*, where *λ**_ML_* and *λ**_MS_* are the edge wavelengths at the long-wavelength side and the short-wavelength side of the filter spectral transmittance, respectively. The central wavelength *λ**_M_* = (*λ**_MS_* + *λ**_ML_*)/2. The definition of the edge width Δ*λ**_M_* is similar to Δ*λ**_C_*. From Equation (7) and the definitions of Δ*λ**_Sep_* and Δ*λ**_M_*, the following equation can be derived:(8)$$\left[{\left(\ln10\right)}^{\frac{1}{a_{M}}}-{\left(-\ln0.9\right)}^{\frac{1}{a_{M}}}\right]\Delta \lambda_{Sep}=2{\left(\ln2\right)}^{\frac{1}{a_{M}}}\Delta \lambda_{M}.$$

Given the values of Δ*λ_Sep_* and Δ*λ_M_*, the value of *a_M_* can be solved from Equation (8). After the value of *a_M_* is solved, the parameter σ*_M_* can be easily calculated by
(9)$$\sigma_{M}=2{\left(\ln2\right)}^{-\frac{1}{a_{M}}}\Delta \lambda_{Sep}$$

The wavelengths *λ_MS_* and *λ_ML_* were taken as the specifications of the magenta filter, where *λ_MS_* = *λ_M_* − Δ*λ_Sep_*/2 and *λ_ML_* = *λ_M_* + Δ*λ_Sep_*/2.

For simplicity, the maximum transmittance of the filters was set to 0.96, i.e., *f_Y_*_0_ = *f_C_*_0_ = *f_M_*_0_ = 0.96; all edge widths for the three filters were set to 30 nm, i.e., Δ*λ**_Y_* = Δ*λ**_C_* = Δ*λ**_M_* = 30 nm. The four edge wavelengths *λ**_C_*, *λ**_Y_*, *λ**_MS_* and *λ**_ML_* were optimized for the minimum mean *E*_Ref_ of the outside samples under the constraints
(10a)$$\lambda_{\mathrm{CamG}}\;\leq \;\lambda_{C}\;\leq \lambda_{\mathrm{CamR}}\;+\;\Delta \lambda_{\mathrm{CamR}}/2,$$
(10b)$$\lambda_{\mathrm{CamB}}\;\leq \;\lambda_{Y}\;\leq \;\lambda_{\mathrm{CamG}}\;+\;\Delta \lambda_{\mathrm{CamG}}/2,$$
(10c)$$\lambda_{\mathrm{CamB}}\;\leq \;\lambda_{\mathrm{MS}}\;\leq \;\lambda_{\mathrm{CamG}},$$
(10d)$$\lambda_{\mathrm{CamG}}\;\leq \;\lambda_{\mathrm{ML}}\;\leq \lambda_{\mathrm{CamR}}\;+\;\Delta \lambda_{\mathrm{CamR}}/2.$$

The constraints are empirical, but reasonable. For example, from Equation (10a), the edge wavelength *λ_C_* of the cyan filter should lie between the wavelengths of the green and red camera channels so that the spectrum amplitudes at short and medium wavelengths are less attenuated. Half the bandwidth of the spectral sensitivity was used as the tolerance for the upper bound in Equation (10a,b,d).

The optimization process consisted of two steps. The first step was to optimize the four edge wavelengths using the Bayesian optimization function “bayesopt” implemented in MATLAB. The objective function is the mean *E*_Ref_ of outside samples. Since Bayesian optimization does not use the derivative of the objective function to find the minimum objective value [29], the second step used the MATLAB optimization function “lsqnonlin” to further optimize the four edge wavelengths, where the result of the first step was taken as the initial trial solution. The function “lsqnonlin” was used because the optimization problem is nonlinear. The optimized edge wavelengths *λ**_C_*, *λ**_Y_*, *λ**_MS_* and *λ**_ML_* are denoted as *λ*_Copt_, *λ*_Yopt_, *λ*_MSopt_ and *λ*_MLopt_, respectively.

The edge wavelengths of the optimized filters for the Nikon D5100 are shown in Table 3. The spectral transmittance of the optimized color filters is shown in Figure 7a. The convex hulls H_A_ and H_RA_ using the optimized filters are shown as blue meshes in Figure 8b and Figure 10, respectively, though they are the same for the case considered.

## 5. Results and Discussion

In this section, in addition to the Nikon D5100, an artificial camera is used as a second camera for comparison. The spectral sensitivities of the artificial camera were assumed to be the CIE 1931 CMFs as shown in Figure 1b. It is called the CMF camera, whose spectral specifications are shown in Table 2. The camera used in the numerical results below is the Nikon D5100 unless otherwise specified.

### 5.1. Interpolation

Table 4 shows the assessment metric statistics for the test samples using the LUT method, where the Nikon D5100 and CMF cameras were used. As can be seen from Table 4, out of the 1066 test samples, about 860 samples were inside samples that can be interpolated. The table also shows the extrapolation results for about 200 outside samples, which are discussed in the next subsection. The assessment metric statistics for the inside samples using the two cameras were about the same except for the color difference Δ*E*_00_. If the spectral sensitivities of a camera are different from the CMFs, the color difference Δ*E*_00_ will be a non-zero value from Equations (1a) and (2). While not zero, most of the inside samples using the Nikon D5100 showed little color difference.

The spectrum reconstructions of the test samples using the PCA and wPCA methods are considered for comparison. In the wPCA method, the *i*-th training sample was multiplied by a weighting factor 1/(Δ*E_i_* + *s*), where Δ*E_i_* is the color difference between the test sample and the *i*-th training sample in CIELAB; *s* is a small-valued constant to avoid division by zero [8]. Weighted training samples were used to derive basis spectra. A camera device model was used to convert RGB signal values into tristimulus values for calculating Δ*E_i_*. A third-order root polynomial regression model (RPRM) was employed and trained using the reference samples [30]. The accuracy of the RPRM was slightly higher than that of the polynomial regression model in this case.

The PCA and wPCA methods were used to reconstruct all test samples using the spectral sensitivities of the Nikon D5100 in Figure 1a. Table 5 shows the assessment metric statistics for the test samples using the PCA and wPCA methods, where the inside samples and outside samples were the same as those using the LUT method. The spectrum reconstruction error using the wPCA method was apparently smaller than that using the PCA method, as expected. Comparing Table 4 with Table 5, we can see that the LUT method outperformed the wPCA method except for *GFC*. Figure 11a,b show the *E*_Ref_ and *GFC* histograms for the 864 inside samples, respectively, where the cases using the LUT, PCA and wPCA methods are shown.

The computation time required for the LUT method is at least two orders of magnitude faster than that required for reconstruction methods using basis spectra that emphasize the relationship between the test and training samples [13]. In this work, the ratio of the computation time required to use the LUT method and wPCA method was 1:80.2, where samples were reconstructed from their signal vector ***C*** to the spectral reflectance vector ***S***_RefRec_ using MATLAB on the Windows 10 platform.

### 5.2. Extrapolation Using the LUT Method Utilizing Optimized ARSs

The color filters optimized as in Section 4.2 were used to create the ARSs in this subsection. The edge wavelengths of the optimized color filters for the Nikon D5100 and CMF cameras are shown in Table 3. The filter spectral transmittance for the Nikon D5100 and CMF cameras is shown in Figure 7a,b, respectively. From Table 4, there were 202 and 203 outside samples for the Nikon D5100 and CMF cameras, respectively. The assessment metric statistics for the outside samples are shown in Table 4. As expected, the mean assessment metrics for the outside samples were worse than those for the inside samples. The assessment metric statistics for the two cameras were about the same except for the color difference Δ*E*_00_. The assessment metric statistics for all samples are also shown in Table 4.

For the Nikon D5100, there were 98, 79, 22 and 3 outside samples that referenced 1, 2, 3 and 4 ARSs, respectively. Figure 12a–f show the reconstructed spectra ***S***_Rec_ using the LUT method for the 2.5G 7/6, 10P 7/8, 2.5R 4/12, 2.5Y 9/4, 10BG 4/8 and 5PB 4/12 color chips, respectively, where their target spectrum ***S***_Reflection_ and neighboring reference spectra are also shown. The case in Figure 12a is an interpolation example for comparison. The cases in Figure 12b–f are extrapolation examples. For the cases in Figure 12b–f, the numbers of referenced ARSs are 1, 2, 2, 3 and 4, respectively. The ARS neighborhoods are indicated in the figures. Neighborhood 3 is the black ARS for the case in Figure 12e. The spectrum was well recovered for the case in Figure 12f, although four ARSs were referenced. The reconstructed spectral reflectance ***S***_RefRec_ for the cases in Figure 12a–f is shown in Figure 13a–f, respectively. RMS errors *E*_Ref_ = 0.004, 0.0223, 0.014, 0.0165, 0.0159 and 0.0149 for the cases in Figure 13a–f, respectively.

### 5.3. Comparison of Extrapolations using Different Methods

#### 5.3.1. PCA and wPCA Methods

Table 5 shows the assessment metric statistics for the 202 outside samples of the Nikon D5100 using the PCA and wPCA methods. As expected, the spectrum reconstruction error using the wPCA method was apparently smaller than that using the PCA method. Comparing Table 4 with Table 5, we can see that the extrapolation using the LUT method utilizing optimized ARSs outperformed the wPCA method. Note that the ratio of good fit *RGF99* was reduced from 0.9803 for the inside samples to 0.7772 for the outside samples when using the wPCA method, i.e., 22.28% of the outside samples had a *GFC* of less than 0.99. When using the LUT method utilizing optimized ARSs, *RGF99* was slightly reduced from 0.9375 for the inside samples to 0.9257 for the outside samples. It was found that when ARSs were included in the training samples of the wPCA method, the extrapolation error did not decrease, but increased further.

Figure 13a–f also show the reconstructed spectral reflectance using the PCA and wPCA methods. The RMS errors *E*_Ref_ = 0.0135, 0.0246, 0.1142, 0.0352, 0.0393 and 0.0366 for the cases using the PCA method in Figure 13a–f, respectively. The RMS errors *E*_Ref_ = 0.0095, 0.0221, 0.0794, 0.03, 0.0297 and 0.0392 for the cases using the wPCA method in Figure 13a–f, respectively.

#### 5.3.2. Nearest Tetrahedron 3D Extrapolation Method

By definition, an outside sample cannot be enclosed by any tetrahedron in the tetrahedral mesh of reference samples in the RGB signal space. However, it can be extrapolated from the nearest tetrahedron [13]. The reference samples of the tetrahedron vertices are used to extrapolate the outside sample, using the same method as interpolation, except that the coefficients in Equation (1a,b) are not restricted to be between 0 and 1. The nearest tetrahedron can be located according to its circumcenter, in-center or centroid. For example, if the locating rule is based on the circumcenter, the nearest tetrahedron is the tetrahedron with the shortest Euclidian distance between its circumcenter and the outside sample.

Table 6 shows the assessment metric statistics for the 202 outside samples of the Nikon D5100 using the nearest tetrahedron 3D extrapolation method. The methods in the table using the locating rules based on the circumcenter, in-center and centroid are designated as NTCC, NTIC and NTCE, respectively. The results using the LUT method utilizing optimized ARSs and the wPCA method shown in Table 4 and Table 5 are also shown in Table 6 for comparison. As can be seen from Table 6, the mean extrapolation error using the NTCC method was much less than that of the NTIC and NTCE methods but was larger than that of the LUT method utilizing optimized ARSs.

#### 5.3.3. LUT Method Utilizing MMSEP Samples

The extrapolation using the LUT method utilizing MMSEP samples is considered. As described in Section 1, eight spectral reflectance functions were constructed so that their color points under D65 illumination were as close as possible to the corners of the RGB signal cube. The eight MMSEP samples were included in the reference sample dataset for extrapolation. The white MMSEP sample and black MMSEP sample are the same as the white ARS and black ARS, respectively. The spectral reflectance functions of the other six MMSEPs are based on the sigmoid function with parameters optimized for the minimum objective function defined in [14].

Table 7 shows the optimized RGB signal values of the MMSEP samples for the Nikon D5100. The RGB signal values were not close to their target values, except for the white and black MMSEP samples. Taking the green MMSEP sample as an example, if the value of its *G* signal is close to 1.0, its *R* and *B* signals will not be small in value because the spectral sensitivities overlap, as shown in Figure 1a. The convex hull of the eight MMSEP samples is shown as a red mesh in Figure 10a,b. The convex hull H_AR_ is smaller in size than the MMSEP convex hull but extends more in the yellow and purple regions. The convex hull H_AR_ can be expanded further if red, green and blue filters are used. The LUT method utilizing MMSEP samples is equivalent to the LUT method utilizing only eight ARSs, where six color filters are used and the white card is the only reflective surface.

The assessment metric statistics for the 202 outside samples of the Nikon D5100 using the LUT method utilizing MMSEP samples are shown in Table 6. There were 148, 46 and 8 outside samples that referenced 1, 2 and 3 MMSEP samples, respectively. As can be seen from Table 6, using the optimized ARSs improved the assessment metrics compared to using MMSEP samples. Figure 13b–f also show the reconstructed spectral reflectance using the LUT method utilizing MMSEP samples, where the RMS errors *E*_Ref_ = 0.0198, 0.0846, 0.0331, 0.0148 and 0.038, respectively. For the cases in Figure 13b–f, the numbers of referenced MMSEP samples are 1, 3, 1, 2 and 1, respectively.

Figure 14a,b show the *E*_Ref_ and *GFC* histograms for the 202 outside samples, respectively, where the cases using the LUT method utilizing optimized ARSs, the wPCA method and the LUT method utilizing MMSEP samples are shown. For extrapolation, the LUT method utilizing optimized ARSs outperformed the wPCA method and the LUT method utilizing MMSEP samples.

### 5.4. Effect of Filter Edge Wavelengths

The spectral sensitivities of a camera can be measured or estimated as described in Section 1. If the measurements or estimates are accurate, the color filters can be optimized using the same method as in Section 4.2. On the other hand, estimates of sensitivity spectral shapes may not be very accurate, but estimates of channel wavelengths may be accurate enough for color filter design. It is found that the specifications of optimized color filters are related to the channel wavelengths. For the Nikon D5100, from Table 2 and Table 3, the optimized edge wavelengths *λ*_Copt_, *λ*_Yopt_, *λ*_MSopt_ and *λ*_MLopt_ are about *λ*_CamR_ +Δ*λ*_CamR_/4, (*λ*_CamC_ + *λ*_CamG_)/2, *λ*_CamC_ and *λ*_CamR_, respectively, where *λ*_CamC_ is an average wavelength and
(11)$$\lambda_{\mathrm{CamC}}\;=\;(\lambda_{\mathrm{CamB}}\;+\;\lambda_{\mathrm{CamG}})/2,$$

These approximate relationships are roughly valid for the CMF camera. Since the channel wavelength is the average wavelength of the spectral sensitivity, the specifications of appropriate color filters could be estimated from less accurate estimates of the spectral sensitivities.

In this subsection, the use of color filters that are not optimized is studied. Deviations of filter specifications from their optimized values are expressed as δ*λ*_C_ = *λ*_C_ − *λ*_Copt_, δ*λ*_Y_ = *λ*_Y_ − *λ*_Yopt_, δ*λ*_MS_ = *λ*_MS_ − *λ*_MSopt_ and δ*λ*_ML_ = *λ*_ML_ − *λ*_MLopt_. It is found that the appropriate range of edge wavelengths can be roughly estimated as
(12a)$$\lambda_{\mathrm{CamY}}\;\leq \;\lambda_{C}\;\leq \;\lambda_{\mathrm{Copt}},$$
(12b)$$\lambda_{\mathrm{CamC}}\;\leq \;\lambda_{Y}\;\leq \;\lambda_{\mathrm{Yopt}},$$
(12c)$$\lambda_{\mathrm{CamB}}\;\leq \;\lambda_{\mathrm{MS}}\;\leq \;\lambda_{\mathrm{MSopt}},$$
(12d)$$\lambda_{\mathrm{CamY}}\;\leq \;\lambda_{\mathrm{ML}}\;\leq \;\lambda_{\mathrm{MLopt}},$$
where *λ*_CamY_ is an average wavelength and
*λ*_CamY_ = (*λ*_CamG_ + *λ*_CamR_)/2.(13)

For the Nikon D5100, *λ*_CamC_ = 498.7 nm and *λ*_CamY_ = 567 nm. The empirical estimates shown in Equation (12a–d) are based on the comparison of Figure 7a,b with Figure 1a,b, respectively, and the tolerance analysis shown below.

The deviation of the edge wavelength from the optimized value results in a change in the convex hull H_RA_. Since *λ*_Copt_ > *λ*_CamR_, increasing positive δ*λ*_C_ will result in an increase in the *R* signal with little change in the *B* and *G* signals, which will cause the ratios *B*/*R* and *G*/*R* to decrease. The decrease in the signal saturation results in a smaller convex hull H_RA_ in the cyan region, and some outside samples may not be extrapolated. Therefore, the upper bound in Equation (12a) is set to the optimized edge wavelength of the cyan filter. The upper bounds in Equation (12b–d) are changed for similar reasons. Since *λ*_CamC_ < *λ*_Yopt_ < *λ*_CamG_, increasing positive δ*λ*_Y_ will result in a greater reduction in the *G* signal, which will cause the ratio *G*/*B* to decrease. Since *λ*_MSopt_ ≈ *λ*_CamC_ and *λ*_MLopt_ ≈ *λ*_CamR_, increasing positive δ*λ*_MS_ and δ*λ*_ML_ will result in a greater increase in the *G* signal and a greater reduction in the *R* signal, respectively, which will cause the ratios *B*/*G* and *R*/*G* to decrease.

Figure 15a–i show the mean RMS error *E*_Ref_ of outside samples versus δ*λ*_MS_ and δ*λ*_ML_, where the values of δ*λ*_C_ and δ*λ*_Y_ are shown in the figures. The camera is the Nikon D5100. In the figures, δ*λ*_C_ = −51.9 nm and −15.5 nm correspond to *λ*_C_ = *λ*_CamY_ and *λ*_CamR_, respectively; δ*λ*_Y_ = −12.1 nm corresponds to *λ*_Y_ = *λ*_CamC_. The white symbols “+” and “x” are the origin (δ*λ*_MS_ = δ*λ*_ML_ = 0 nm) and the point of the minimum mean *E*_Ref_ in each figure, respectively. The values of the mean *E*_Ref_ at the origin and at the point of the minimum mean *E*_Ref_ are shown in Table 8 and Table 9, respectively, for the cases in Figure 15a–i. In the two tables, the corresponding filter edge wavelength deviations and the ratio *RGF99* are also shown.

At the origin in Figure 15a–i, all outside samples can be extrapolated. Away from the origin, the red dotted line represents the boundary where at least one outside sample cannot be extrapolated. Beyond the boundary, the mean *E*_Ref_ of outside samples that can be extrapolated is shown. The white dotted line in the figure represents the contour of the mean *E*_Ref_ = 0.0213, which is the value obtained using the wPCA method. Using color filters that meet the specifications within both the red and white dotted lines, all outside samples can be extrapolated while keeping the mean *E*_Ref_ < 0.0213. For the cases with δ*λ*_C_ = 0 in Figure 15c,f,i, the area enclosed by the red and white dotted lines is small because some cyan outside samples cannot be extrapolated. They can be extrapolated by using a blue-shift cyan filter, i.e., δ*λ*_C_ < 0, but will increase the extrapolation error. When δ*λ*_C_ = −9.0 nm, the red dotted line is about the same as in the cases of δ*λ*_C_ = −15.5 nm in Figure 15b,e,h. Therefore, if a larger edge wavelength tolerance is required, a blue-shift cyan filter is preferred. For such a requirement, the upper bound *λ*_Copt_ in Equation (12a) can be replaced by *λ*_CamR_.

The point of (δ*λ*_ML_, δ*λ*_MS_) = (−41.1 nm, −33 nm) in Figure 15g is the lower bound case in Equation (12a–d), where *λ*_C_ = *λ*_CamY_, *λ*_Y_ = *λ*_CamC_, *λ*_MS_ = *λ*_CamB_ and *λ*_ML_ = *λ*_CamY_. Since all filter edge wavelengths are blue-shifted, this case is called the blue-shift lower bound (BSLB) case. In contrast, the upper bound case in Equation (12a–d) is the optimized case at the origin in Figure 15c. The assessment metric statistics of the BSLB case are shown in Table 6, where the mean *E*_Ref_ = 0.019 and *RGF99* = 0.8416. As can be seen from Table 6, the assessment metric statistics of the BSLB case were worse than those of the optimized case, but better than those of the wPCA and NTCC methods. Figure 13b–f also show the reconstructed spectral reflectance of the BSLB case, where the RMS errors *E*_Ref_ = 0.0187, 0.0469, 0.0203, 0.0274 and 0.0482, respectively.

Figure 16a,b show the *E*_Ref_ and *GFC* histograms, respectively, for the 202 outside samples of the BSLB case. Also shown are the results of the cases using the LUT method utilizing optimized ARSs and the LUT method utilizing MMSEP samples for comparison. As can be seen from the two figures and Table 6, for the RMS error *E*_Ref_, the unoptimized BSLB case was slightly better than the case including MMSEP samples, but for the goodness-of-fit coefficient *GFC*, the case including MMSEP samples was slightly better. The extrapolation performance of the two cases is comparable. Note that it is easy to design better color filters than the BSLB case for extrapolation.

## 6. Conclusions

The reconstruction of spectral reflectance using the LUT method to interpolate camera RGB signals was investigated. Using the LUT method has the advantages of a high accuracy, saving computation time and eliminating the need to use the camera spectral sensitivity functions. The disadvantage of this method is that it cannot interpolate samples outside the convex hull of the reference samples in the RGB signal space. The outside samples can be extrapolated by using the method based on basis spectra, but it has two disadvantages: (1) accurate camera spectral sensitivity functions are required; (2) the calculation of the algorithm with a low spectrum reconstruction error is time-consuming. This paper proposed the LUT method utilizing auxiliary reference samples for extrapolating outside samples. The auxiliary reference samples were created by using reference samples and color filters so that the convex hull of the reference samples and auxiliary reference samples can enclose the outside samples in the RGB signal space. Therefore, outside samples can be extrapolated by using the LUT method utilizing auxiliary reference samples.

The proposed method was tested with Munsell color chips as examples of reflective surfaces. The Nikon D5100 camera was taken as an example camera. The method to create auxiliary reference samples was described. Cyan, yellow and magenta filters were used in this study. The optimized design of the three filters was presented. The results show that the mean extrapolation error using the proposed method was smaller than that of the weighted PCA method. The specifications for the optimized color filters mainly depend on the average wavelengths of the camera spectral sensitivities. The appropriate range of color filter edge wavelengths was shown. The design of color filters may not require accurate measurement or estimation of the camera spectral sensitivities. Therefore, the proposed method is feasible for overcoming the extrapolation problem. It was also shown that the ratio of computation time required to use the LUT method and the wPCA method was 1:80.2.

Since only one commercially available camera was considered, further studies are required for cameras with other sensitivity characteristics. Further research should use more than three color filters to expand the convex hull of the reference samples and auxiliary reference samples, and further reduce the extrapolation error where possible. Studies implementing the proposed method for field application will be published in the future.

## Figures and Tables

**Figure 1 sensors-22-04923-f001:**
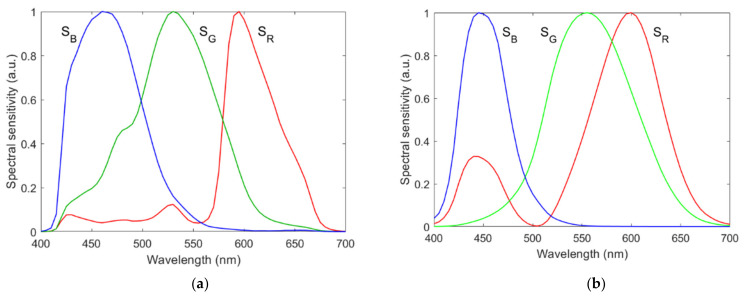
Spectral sensitivities of (**a**) the Nikon D5100 and (**b**) the CMF camera. Spectral sensitivities of the red, green and blue signal channels are denoted as **S**_R_, **S**_G_ and **S**_B_, respectively.

**Figure 2 sensors-22-04923-f002:**
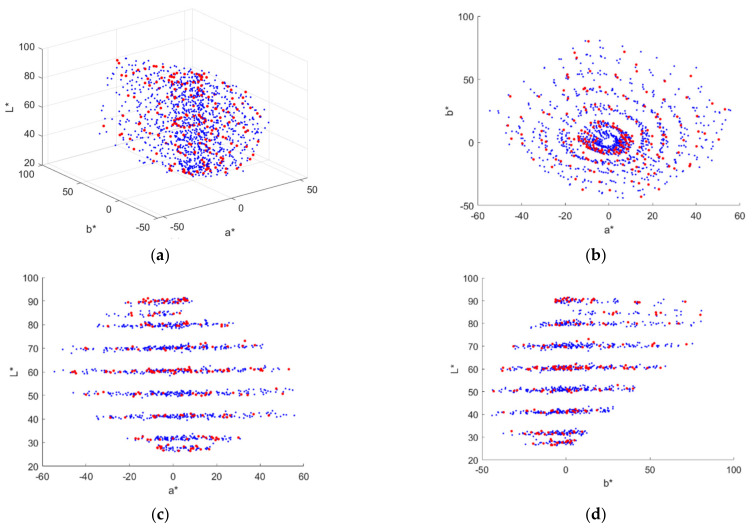
(**a**) Color points of the reflection light from Munsell color chips in CIELAB. (**b**–**d**) show the color points projected on the a*b* plane, a*L* plane and b*L* plane, respectively. Reference samples and test samples are shown as red and blue dots, respectively. The illuminant is D65.

**Figure 3 sensors-22-04923-f003:**
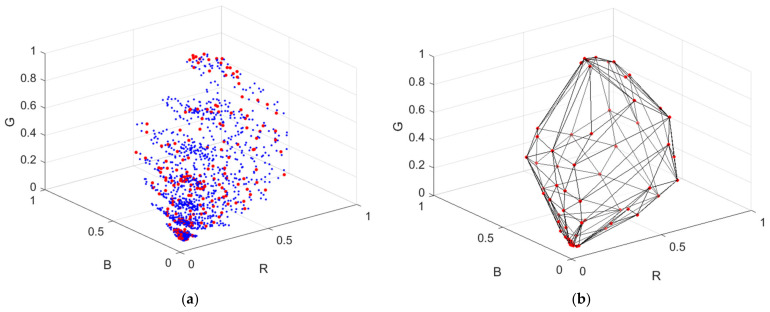
(**a**) Color points of the reflection light from Munsell color chips in the RGB signal space using the Nikon D5100, where reference samples and test samples are shown as red and blue dots, respectively. The illuminant is D65. (**b**) Convex hull of reference samples H_R_.

**Figure 4 sensors-22-04923-f004:**
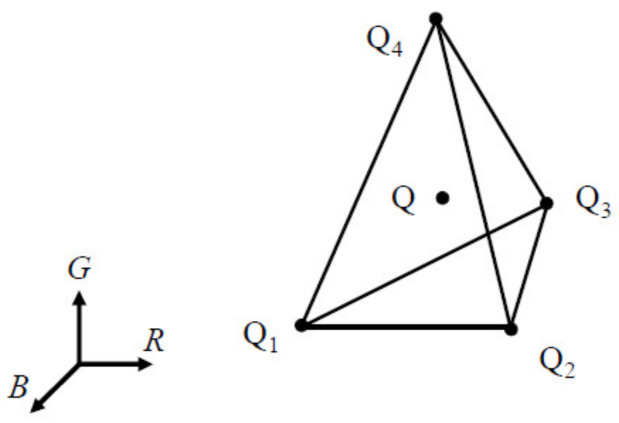
Schematic diagram showing a color point Q enclosed by a tetrahedron with vertices Q_1_, Q_2_, Q_3_ and Q_4_ in the RGB signal space.

**Figure 5 sensors-22-04923-f005:**
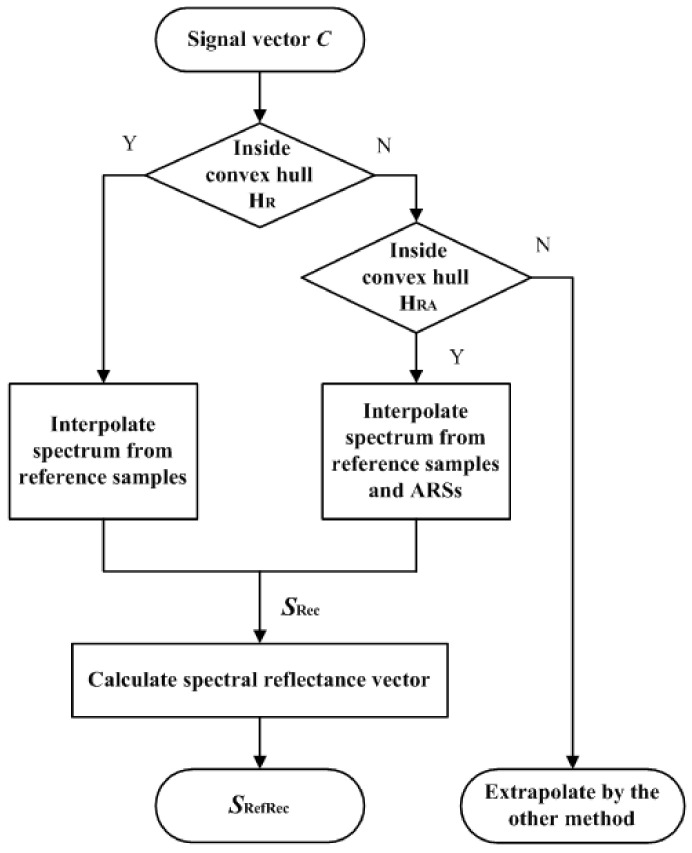
Flow chart for reconstructing the spectral reflectance vector ***S***_RefRec_ from the signal vector ***C***. Refer to Section 3.2 for details.

**Figure 6 sensors-22-04923-f006:**
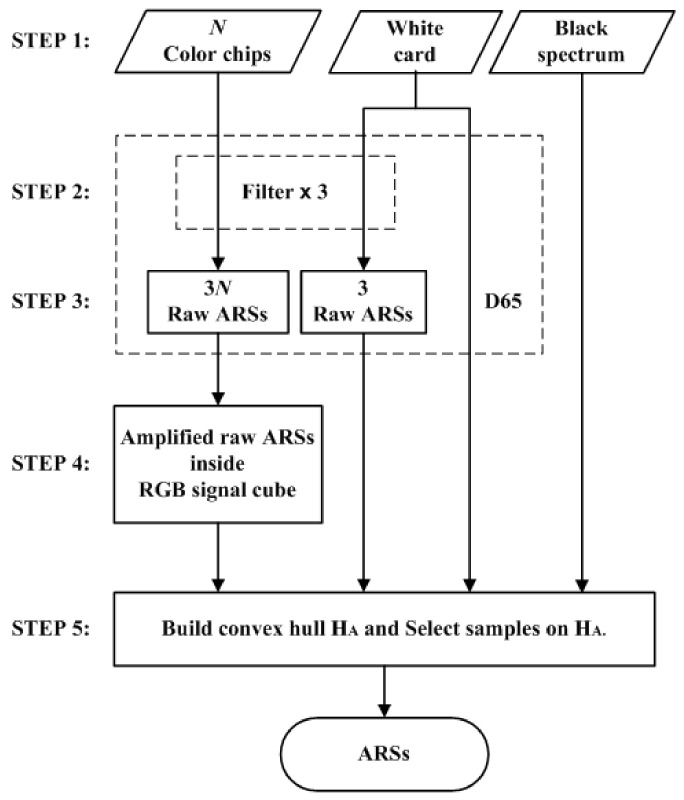
Flow chart for creating auxiliary reference samples (ARSs). Refer to Section 4.1 for details.

**Figure 7 sensors-22-04923-f007:**
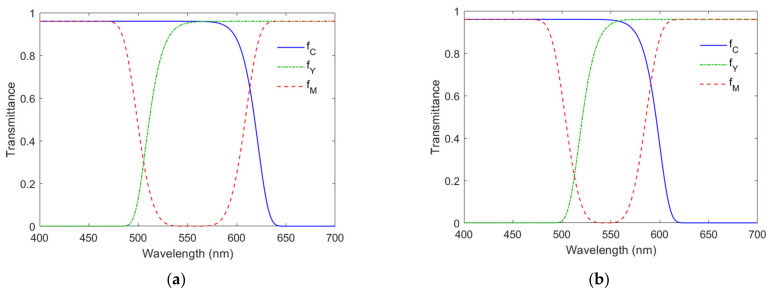
Spectral transmittance of the color filters optimized for (**a**) the Nikon D5100 and (**b**) the CMF camera. The spectral transmittance of the cyan, yellow and magenta filters is denoted as *f*_C_, *f*_Y_ and *f*_M_, respectively.

**Figure 8 sensors-22-04923-f008:**
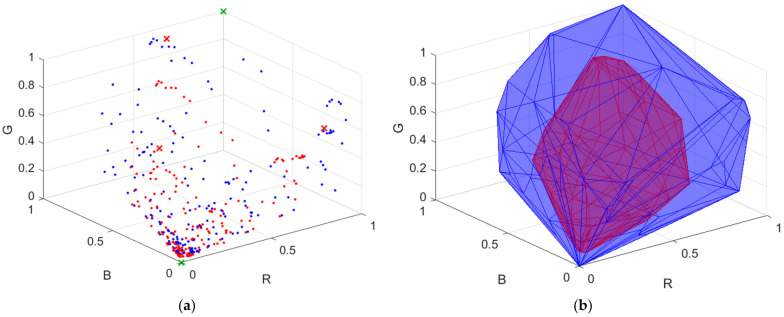
(**a**) Color points of raw ARSs and amplified raw ARSs shown as red and blue dots, respectively, in the RGB signal space. The raw ARSs from the white card are shown as red crosses. The white and black ARSs are shown as green crosses. Color filters optimized for the Nikon D5100 are used. (**b**) The convex hull H_A_ of the ARSs and the convex hull H_R_ of the reference samples are shown as blue and red meshes, respectively. Figure 3b shows the same H_R_.

**Figure 9 sensors-22-04923-f009:**
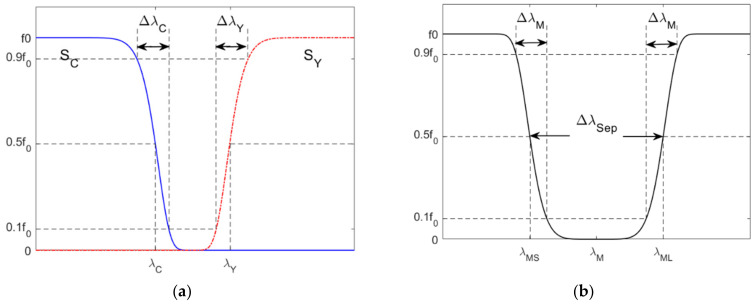
Schematic diagrams showing the specification definitions of the (**a**) cyan and yellow color filters, and (**b**) magenta filter. Refer to Section 4.2 for details.

**Figure 10 sensors-22-04923-f010:**
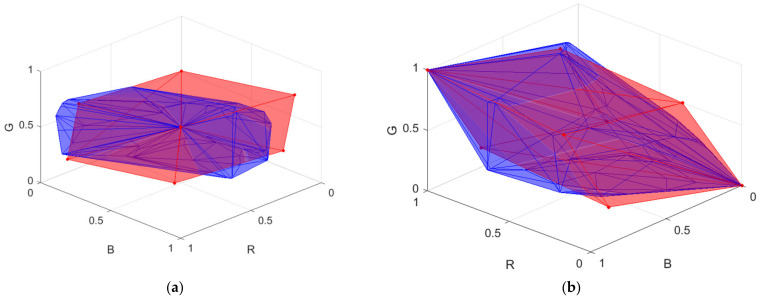
(**a**) The convex hull H_RA_ of the reference samples and ARSs and the convex hull of the MMSEP samples are shown as blue and red meshes, respectively. The ARSs are the same as in Figure 3b, where optimized color filters are used. The convex hull H_RA_ is the same as the convex hull H_A_ in Figure 8b, but the viewing angle is different. (**b**) is the same as (**a**), except it rotates 90° clockwise along the *G* axis.

**Figure 11 sensors-22-04923-f011:**
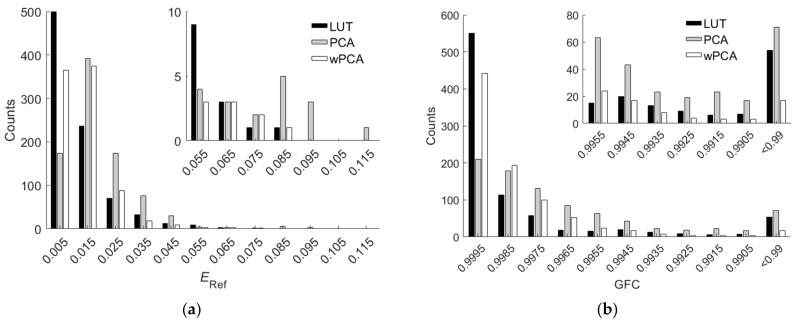
(**a**) *E*_Ref_ and (**b**) *GFC* histograms for the 864 inside samples of the cases using the LUT, PCA and wPCA methods. The camera is the Nikon D5100. The insets show enlarged parts. In (**b**), all the counts in the “<0.99” slot have *GFC* < 0.99.

**Figure 12 sensors-22-04923-f012:**
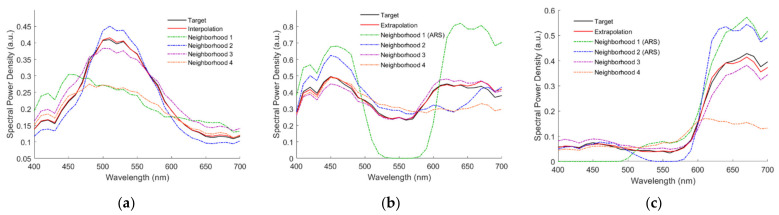
Target spectrum ***S***_Reflection_, reconstructed spectra ***S***_Rec_ and neighboring reference spectra using the LUT method and optimized color filters. Munsell annotations of the color chips are (**a**) 2.5G 7/6, (**b**) 10P 7/8, (**c**) 2.5R 4/12, (**d**) 2.5Y 9/4, (**e**) 10BG 4/8 and (**f**) 5PB 4/12.

**Figure 13 sensors-22-04923-f013:**
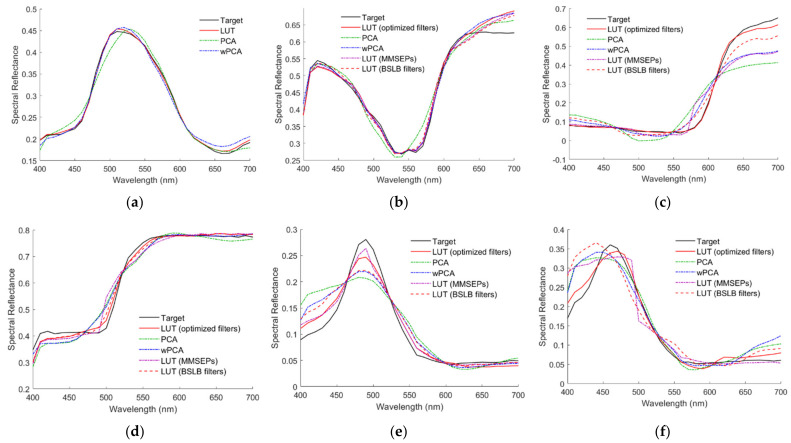
(**a**–**f**) show the target spectral reflectance ***S***_Ref_ and reconstructed spectral reflectance ***S***_RefRec_ for the cases in Figure 12a–f, respectively. The results using the other reconstruction methods are also shown. Refer to Section 5.1, Section 5.2, Section 5.3 and Section 5.4 for details.

**Figure 14 sensors-22-04923-f014:**
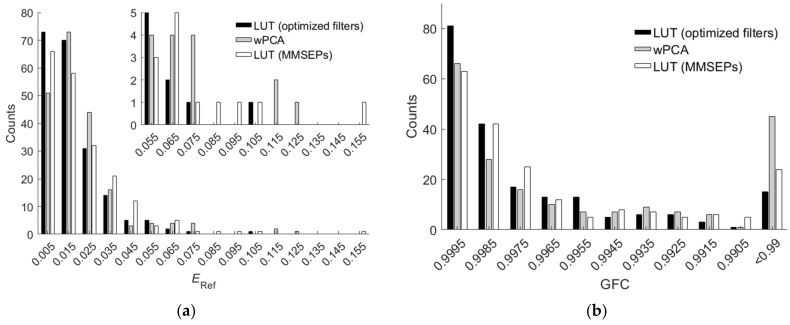
(**a**) *E*_Ref_ and (**b**) *GFC* histograms for the 202 outside samples of the cases using the LUT and wPCA methods. The results of the cases using the LUT method utilizing optimized ARSs and the LUT method utilizing MMSEP samples are shown. In (**a**), the inset shows an enlarged part. In (**b**), all the counts in the “<0.99” slot have *GFC* < 0.99.

**Figure 15 sensors-22-04923-f015:**
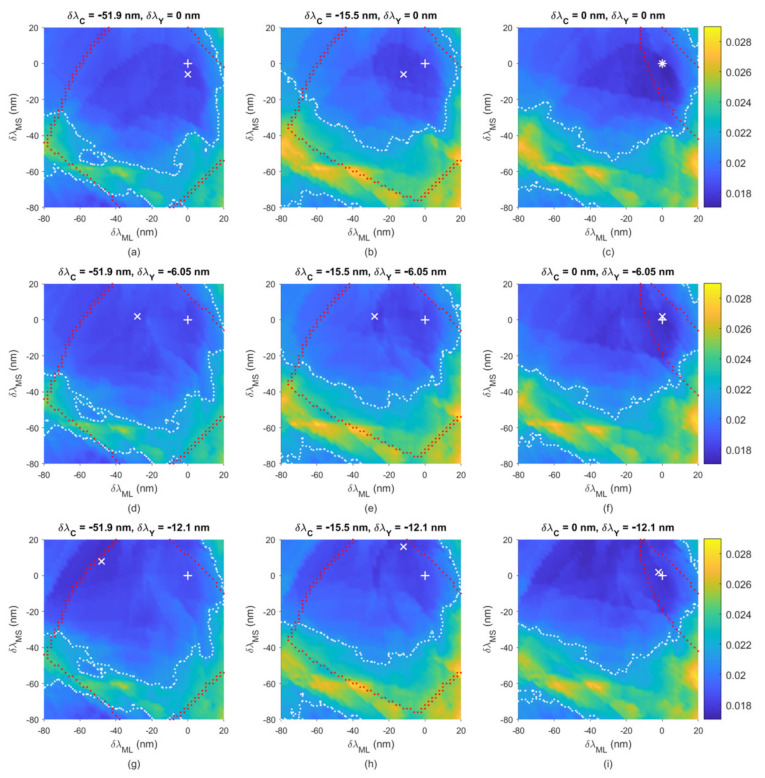
Mean *E*_Ref_ of 202 outside samples versus δ*λ*_M_ and δ*λ*_Sep_ for the Nikon D5100, where the values of δ*λ*_C_ and δ*λ*_Y_ are shown in (**a**–**i**). The white symbols “+” and “×” are the origin and the point of the minimum mean *E*_Ref_ in the figure, respectively. The red dotted line is the boundary where at least one outside sample cannot be extrapolated. Beyond the boundary, the mean *E*_Ref_ of outside samples that can be extrapolated is shown. The white dotted line is the contour of *E*_Ref_ = 0.0213.

**Figure 16 sensors-22-04923-f016:**
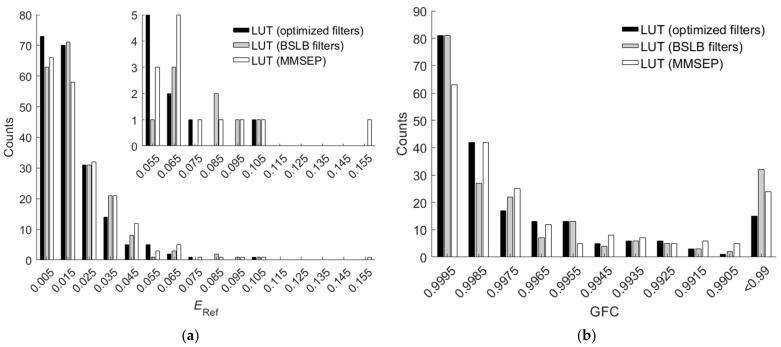
(**a**) *E*_Ref_ and (**b**) *GFC* histograms for the 202 outside samples of the BSLB case. Also shown are the results of the cases using the LUT method utilizing optimized ARSs and the LUT method utilizing MMSEP samples for comparison. In (**a**), the inset shows an enlarged part. In (**b**), all the counts in the “<0.99” slot have *GFC* < 0.99.

**Table 1 sensors-22-04923-t001:** Abbreviation list.

Abbreviation	Definition
ARS	Auxiliary Reference Sample
BSLB	Blue-Shift Lower Bound
CMF	Color Matching Function
FWHM	Full Width at Half Maximum
GFC	Goodness-of-Fit Coefficient
LUT	Look-Up Table
MAX	Maximum
MIN	Minimum
MMSEP	Model-based Metameric Spectra of Extreme Point
NMT	Non-negative Matrix Transformation
NTCC	Nearest Tetrahedron based on Circumcenter
NTCE	Nearest Tetrahedron based on Centroid
NTIC	Nearest Tetrahedron based on In-Center
PC50	50th Percentile
PC98	98th Percentile
PCA	Principal Component Analysis
RGF99	Ratio of Good Fit. (The ratio of samples with *GFC* > 0.99.)
RMS	Root Mean Square
RPRM	Root Polynomial Regression Model
SCI	Spectral Comparison Index
wPCA	Weighted Principal Component Analysis

**Table 2 sensors-22-04923-t002:** Spectral specifications of the Nikon D5100 and CMF cameras.

Specification	Channel Wavelength (nm)			FWHM Spectral Width (nm)		
Channel	B (*λ*_CamB_)	G (*λ*_CamG_)	R (*λ*_CamR_)	B (Δ*λ*_CamB_)	G (Δ*λ*_CamG_)	R (Δ*λ*_CamR_)
D5100	466.7	530.7	603.4	80.1	88.3	55.8
CMF	452.2	559.2	588.1	55.2	100.4	79.4

**Table 3 sensors-22-04923-t003:** Edge wavelengths of the optimized color filters for the Nikon D5100 and CMF cameras.

	*λ*_Copt_ (nm)	*λ*_Yopt_ (nm)	*λ*_MSopt_ (nm)	*λ*_MLopt_ (nm)
D5100	618.9	510.8	499.7	608.1
CMF	596.7	520.5	504.0	585.4

**Table 4 sensors-22-04923-t004:** Assessment metric statistics for test samples using the LUT method and optimized color filters. The Nikon D5100 and CMF cameras were used.

Metric	Camera	Nikon D5100			CMF		
	Sample	All	Inside	Outside	All	Inside	Outside
	No.	1066	864	202	1066	863	203
*E* _Ref_	mean *μ*	0.0129	0.0120	0.0171	0.0132	0.0123	0.0169
	std *σ*	0.0118	0.0107	0.0150	0.0116	0.0103	0.0152
	*PC50*	0.0091	0.0087	0.0132	0.0099	0.0094	0.0132
	*PC98*	0.0509	0.0485	0.0599	0.0494	0.0444	0.0650
	*MAX*	0.1078	0.0859	0.1078	0.1111	0.0816	0.1111
*GFC*	mean *μ*	0.9972	0.9974	0.9962	0.9972	0.9974	0.9960
	std *σ*	0.0074	0.0071	0.0084	0.0063	0.0054	0.0090
	PC50	0.9993	0.9994	0.9986	0.9993	0.9994	0.9986
	*MIN*	0.9000	0.9000	0.9193	0.9161	0.9457	0.9161
	*RGF99*	0.9353	0.9375	0.9257	0.9203	0.9212	0.9163
Δ*E*_00_	mean *μ*	0.4244	0.4239	0.4262	0.0000	0.0000	0.0000
	std *σ*	0.4115	0.4182	0.3827	0.0000	0.0000	0.0000
	*PC50*	0.2823	0.2795	0.3015	0.0000	0.0000	0.0000
	*PC98*	1.6842	1.6900	1.6402	0.0000	0.0000	0.0000
	*MAX*	2.5918	2.5918	1.8962	0.0000	0.0000	0.0000
*SCI*	mean *μ*	4.1102	3.7503	5.6495	4.1869	3.8632	5.5631
	std *σ*	3.1802	2.9266	3.7252	3.0695	2.8885	3.4233
	*PC50*	3.1484	2.9310	4.6951	3.3348	3.1732	4.8976
	*PC98*	13.4611	12.1239	15.0412	12.4129	11.7579	14.3999
	*MAX*	25.2299	25.2299	21.9370	23.8934	23.8934	15.7186

**Table 5 sensors-22-04923-t005:** Assessment metric statistics for test samples using the PCA and wPCA methods. The camera is the Nikon D5100.

Metric	Method	PCA			wPCA		
	Sample	All	Inside	Outside	All	Inside	Outside
	No.	1066	864	202	1066	864	202
*E* _Ref_	mean *μ*	0.0221	0.0193	0.0341	0.0147	0.0131	0.0213
	std *σ*	0.0168	0.0128	0.0247	0.0122	0.0092	0.0194
	*PC50*	0.0173	0.0160	0.0276	0.0121	0.0114	0.0155
	*PC98*	0.0817	0.0515	0.1152	0.0565	0.0402	0.0774
	*MAX*	0.1442	0.1180	0.1442	0.1255	0.0894	0.1255
*GFC*	mean *μ*	0.9940	0.9958	0.9860	0.9972	0.9982	0.9931
	std *σ*	0.0101	0.0072	0.0155	0.0062	0.0031	0.0118
	PC50	0.9974	0.9977	0.9892	0.9990	0.9990	0.9976
	*MIN*	0.8858	0.8982	0.8858	0.8921	0.9444	0.8921
	*RGF99*	0.8349	0.9178	0.4802	0.9418	0.9803	0.7772
Δ*E*_00_	mean *μ*	0.8261	0.6970	1.3780	0.5017	0.4318	0.8011
	std *σ*	0.6202	0.4163	0.9572	0.4650	0.3099	0.7887
	*PC50*	0.7003	0.6488	1.1215	0.3793	0.3600	0.5094
	*PC98*	2.8667	1.7963	3.6855	2.2116	1.2505	3.0850
	*MAX*	4.3546	3.0765	4.3546	3.3029	2.3674	3.3029
*SCI*	mean *μ*	7.8531	6.3217	14.4032	4.7942	3.9033	8.6050
	std *σ*	6.5329	4.2839	9.7024	4.3745	2.7381	7.1555
	*PC50*	5.8291	5.0862	12.0428	3.4185	3.1503	6.3268
	*PC98*	27.2764	19.6718	38.8951	19.1506	11.8655	31.3431
	*MAX*	55.4331	27.4239	55.4331	35.2093	26.8467	35.2093

**Table 6 sensors-22-04923-t006:** Assessment metric statistics for the 202 outside samples of the Nikon D5100 using the NTCC method, NTIC method, NTCE method, LUT method utilizing MMSEP samples and LUT method utilizing ARSs in the BSLB case (ARS(BSLB)). Also shown are the cases using the LUT method utilizing optimized ARSs (ARS(Opt)) and the wPCA method for comparison.

Metric	Statistics	NTCC	NTIC	NTCE	MMSEP	ARS (BSLB)	ARS (Opt)	wPCA
*E* _Ref_	mean *μ*	0.0258	0.0393	0.0470	0.0212	0.0190	0.0171	0.0213
	std *σ*	0.0225	0.0600	0.0712	0.0201	0.0164	0.0150	0.0194
	*PC50*	0.0196	0.0173	0.0193	0.0164	0.0154	0.0132	0.0155
	*PC98*	0.0967	0.2493	0.3295	0.0819	0.0728	0.0599	0.0774
	*MAX*	0.1279	0.4292	0.4292	0.1531	0.1003	0.1078	0.1255
*GFC*	mean *μ*	0.9856	0.9703	0.9617	0.9951	0.9948	0.9962	0.9931
	std *σ*	0.0313	0.0878	0.0981	0.0090	0.0093	0.0084	0.0118
	PC50	0.9971	0.9965	0.9961	0.9981	0.9982	0.9986	0.9976
	*MIN*	0.8355	0.3718	0.3718	0.9356	0.9376	0.9193	0.8921
	*RGF99*	0.7475	0.6931	0.6485	0.8812	0.8416	0.9257	0.7772
Δ*E*_00_	mean *μ*	0.7198	1.0659	1.7758	0.5900	0.6847	0.4262	0.8011
	std *σ*	0.6564	1.6787	4.3453	0.5657	0.6437	0.3827	0.7887
	*PC50*	0.5217	0.5207	0.5629	0.3419	0.4782	0.3015	0.5094
	*PC98*	2.8621	6.8300	21.3735	2.1651	2.5164	1.6402	3.0850
	*MAX*	3.4551	15.3924	32.8994	2.8408	3.2615	1.8962	3.3029
*SCI*	mean *μ*	8.7137	12.5158	16.5762	6.9223	7.3756	5.6495	8.6050
	std *σ*	6.0169	16.4821	26.2679	5.5522	5.7411	3.7252	7.1555
	*PC50*	7.5725	7.1176	7.6722	5.7158	5.4641	4.6951	6.3268
	*PC98*	25.7363	61.6082	130.5460	23.4707	22.2341	15.0412	31.3431
	*MAX*	30.4515	135.4394	157.6624	38.2342	32.9033	21.9370	35.2093

**Table 7 sensors-22-04923-t007:** Target and optimized RGB signal values of MMSEP samples for the Nikon D5100.

	Target			Optimized		
Signal	R	G	B	R	G	B
Red	1	0	0	0.8174	0.1221	0.008
Green	0	1	0	0.1874	0.6941	0.19
Blue	0	0	1	0.0736	0.2255	0.8014
Cyan	0	1	1	0.1771	0.8719	0.9863
Magenta	1	0	1	0.8506	0.3245	0.8051
Yellow	1	1	0	0.9261	0.7720	0.1972
White	1	1	1	1	1	1
Black	0	0	0	0	0	0

**Table 8 sensors-22-04923-t008:** Filter edge wavelength deviations and the mean RMS error *E*_Ref_ of outside samples at the origin in Figure 15a–i. The ratio *RGF99* of outside samples is also shown.

Figure 15	δ*λ*_C_ (nm)	δ*λ*_Y_ (nm)	δ*λ*_MS_ (nm)	δ*λ*_ML_ (nm)	*Mean E* _Ref_	*RGF99*
(**a**)	−51.9	0	0	0	0.0181	0.8614
(**b**)	−15.5	0	0	0	0.0182	0.8713
(**c**)	0	0	0	0	0.0171	0.9257
(**d**)	−51.9	−6.05	0	0	0.0182	0.8564
(**e**)	−15.5	−6.05	0	0	0.0183	0.8762
(**f**)	0	−6.05	0	0	0.0173	0.9208
(**g**)	−51.9	−12.1	0	0	0.0187	0.8614
(**h**)	−15.5	−12.1	0	0	0.0185	0.8812
(**i**)	0	−12.1	0	0	0.0176	0.9257

**Table 9 sensors-22-04923-t009:** The minimum mean RMS error *E*_Ref_ of outside samples in Figure 15a–i and corresponding filter edge wavelength deviations. The ratio *RGF99* of outside samples is also shown.

Figure 15	δ*λ*_C_ (nm)	δ*λ*_Y_ (nm)	δ*λ*_MS_ (nm)	δ*λ*_ML_ (nm)	*Mean E* _Ref_	*RGF99*
(**a**)	−51.9	0	−6	0	0.018	0.8614
(**b**)	−15.5	0	−6	−12	0.0178	0.8911
(**c**)	0	0	0	0	0.0171	0.9257
(**d**)	−51.9	−6.05	2	−28	0.0181	0.8713
(**e**)	−15.5	−6.05	2	−28	0.0179	0.9109
(**f**)	0	−6.05	2	0	0.0173	0.9208
(**g**)	−51.9	−12.1	8	−48	0.0178	0.8663
(**h**)	−15.5	−12.1	16	−12	0.0177	0.9158
(**i**)	0	−12.1	2	−2	0.175	0.9307

## Data Availability

The data presented in this study are openly available in [16,23].

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
