# Peer review of "Auxiliary Reference Samples for Extrapolating Spectral Reflectance from Camera RGB Signals"

_sensors, 2022, doi:10.3390/s22134923_

Round 1

Reviewer 1 Report

The text contains a large number of abbreviations and designations. This makes reading a bit difficult. Perhaps a list of abbreviations used should be added

It is suspected that LUT tables based on the measurement of the Munsell atlas will only predict samples with the least error from the Munsell atlas itself. Were the results checked on other external samples?  

The algorithm by which the reference vertices of the tetrahedron in Fig. 4 are selected is not described. It seems that tetrahedra can arise, the vertices of which lie almost in a plane. In this case, one of the coefficients in formula 1 will be close to zero, and such a solution will obviously be unstable. Has this issue been considered?  

Reviewer 2 Report

I think the term ‘optimized’ in ‘optimized color filters’ is not appropriate because the filters are empirically decided in the paper, not obtained by solving a mathematical optimization problem.

Section 4.2 elaborates the formulations of the selected filters and specifies the parameters. Are the filters actually manufactured using the formula and parameters? If yes, the difference between theoretic and actual filters should be given; If no, I do not think such a detailed description is necessary.

Reviewer 3 Report

This paper proposes the interpolation method including the auxiliary reference samples for extrapolating spectral reflectance from camera RGB signals. This paper have showed the proper experimental results and have described adequately the proposed motivation and method. It is suitable to publish this manuscript if this paper has the easier explanation each equation to read easily the reader.

Reviewer 4 Report

This paper presents a color interpolation method based on auxiliary reference samples (ARSs) to extrapolate the outside samples of the convex hull of reference samples. Three optimized color filers are designed to create the required ARSs. Experimental results show that the proposed method produces smaller extrapolation errors than the traditional weighted PCA method. I have some suggestions below.

1.      The topic of this paper is related to color gamut mapping in color science; however, the authors miss the discussion of color gamut mapping techniques relevant to this paper.

2.      The proposed LUT interpolation method has been widely used in the field of color gamut mapping; for example, Lee et al. already published a color gamut mapping method using tetrahedral interpolation for color reproduction enhancement about 10 years ago. (H.-S. Lee et al., A real-time color gamut mapping using tetrahedral interpolation for digital TV color reproduction enhancement, IEEE Trans. on Consumer Electronics, vol. 55, no. 2, 2009). Therefore, the authors should explain the advantages of the proposed interpolation method over traditional gamut mapping interpolation methods.

3.      There are already gamut extension algorithms that can map a small color gamut to a large one; for example, L. Xu et al., Color gamut mapping between small and large color gamuts: part II, gamut extension, Optics Express, vol. 28, no. 13, 2018. The authors should compare their method with these existing gamut extension algorithms.

Round 2

Reviewer 4 Report

I have no further comments.